# Learning to Route LLMs with Confidence Tokens

**Yu-Neng Chuang** [* 1]  **Prathusha K. Sarma** [2]  **Parikshit Gopalan** [2]
**John Boccio** [2]  **Sara Bolouki** [2]  **Xia Hu** [1]  **Helen Zhou** [2]

## Abstract

Large language models (LLMs) have demonstrated impressive performance on several tasks and are increasingly deployed in real-world applications. However, especially in high-stakes settings, it becomes vital to know when the output of an LLM may be unreliable. Depending on whether an answer is trustworthy, a system can then choose to route the question to another expert, or otherwise fall back on a safe default behavior. In this work, we study the extent to which LLMs can reliably indicate confidence in their answers, and how this notion of confidence can translate into downstream accuracy gains. We propose Self-Reflection with Error-based Feedback (Self-REF), a lightweight training strategy to teach LLMs to express confidence in whether their answers are correct in a reliable manner. Self-REF introduces confidence tokens into the LLM, from which a confidence score can be extracted. Compared to conventional approaches such as verbalizing confidence and examining token probabilities, we demonstrate empirically that confidence tokens show significant improvements in downstream routing and rejection learning tasks.

## 1. Introduction

Recent years have seen an explosive growth in the deployment of large language models, in forms ranging from helpful conversational agents (Achiam et al., 2023; Touvron et al., 2023; Yang et al., 2024), to frameworks for automating software workflows (Harrison, 2022; Wu et al., 2023), to tools tailored to domain-specific tasks. LLMs have been used to summarize doctor-patient interactions (Krishna et al., 2021; Jeong et al., 2024), teach students (Schmucker et al.,

2023; Xiao et al., 2023), and provide customer support (Kolasani, 2023). As LLMs are given more agency in settings of increasing consequence, it becomes crucial to know when an output is reliable. Given knowledge of the trustworthiness of an output, systems can proactively prevent faulty behavior by seeking a second opinion (e.g. by routing to a more costly LLM), or choosing to abstain from answering (e.g. instead defaulting to a safe behavior).

However, state-of-the-art LLMs face challenges with providing an accurate estimate of confidence in whether a prediction is correct. Several works identify that token probabilities (derived from a softmax over logits) are not well-aligned, or calibrated, with the actual probabilities of correctness (Huang et al., 2023; Detommaso et al., 2024; Wightman et al., 2023). Others have found that modifying the prompt by asking the LLM to verbalize its confidence yields slightly better results (Tian et al., 2023; Turpin et al., 2024; Mahaut et al., 2024); however, these results may be subject to the dataset and prompt engineering, often leading to unstable or unreliable results. Since LLMs are typically trained using a cross-entropy loss, they can overfit on accuracy rather than calibration, often leading to overconfidence and misalignment with real-world distributions. Thus, a natural question arises: *how can we train LLMs to accurately estimate their own confidence levels, while preserving their performance on tasks of interest?*

To answer this question, we introduce **Self-R**eflection with **E**rror-based **F**eedback (**Self-REF**), a lightweight framework for fine-tuning LLMs to accurately assess confidence in their predictions while preserving their performance on relevant downstream tasks. Self-REF integrates naturally with any LLM backbone, allowing the model to condition its assessment of confidence on its generated outputs. It consists of three steps: *(i) Confidence token annotation*: the training dataset is generated by labeling each instance with confidence tokens based on the correctness of the base LLM. When the base LLM provides correct responses to the input text, we augment the instances with confident (`<CN>`) tokens. Conversely, if it responds incorrectly, we augment the instances with unconfident (`<UN>`) tokens. *(ii) Self-REF fine-tuning*: The base LLM is fine-tuned on the augmented training data, so that one would expect a confidence token to be generated following its predicted output. *(iii) Confidence*

---

[*]Work done during the internship at Apple. [1]Rice University, Houston, TX, USA [2]Apple, Inc., Cupertino, CA, USA. Correspondence to: Yu-Neng Chuang <ynchuang@rice.edu>, Helen Zhou <helen_l_zhou@apple.com>.

*Proceedings of the $42^{nd}$ International Conference on Machine Learning*, Vancouver, Canada. PMLR 267, 2025. Copyright 2025 by the author(s).

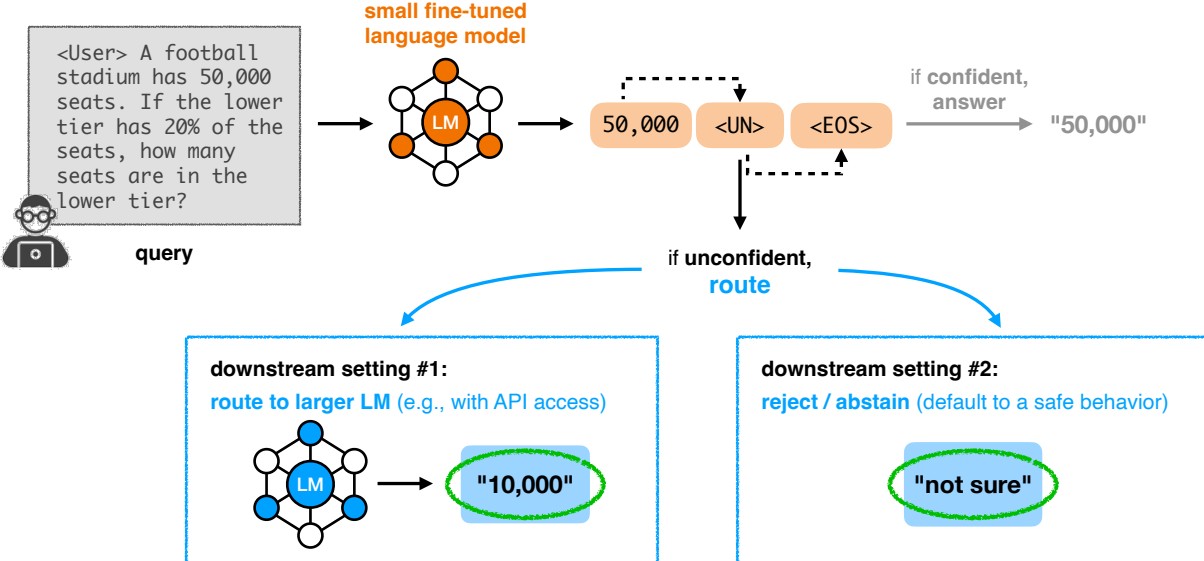

Figure 1: Self-REF attaches a confidence token to each prediction. Routing proceeds based on the confidence.

*score extraction*: To obtain a continuous confidence score, one computes the probability of the <CN> token, normalized by the sum of the probabilities of the <UN> and <CN> tokens.

Given estimates of confidence, a natural follow-up question is how to *utilize* such estimates to yield desirable system-level behavior. Often, one may have limited computational resources and only be able to locally fine-tune smaller models. However, one may have API access to a larger model with higher base performance for some cost. In this work, we focus on two downstream tasks that utilize this notion of confidence for practical applications: (1) **routing**, where queries with low confidence are routed to stronger but more costly LLMs, and (2) **rejection**, where the model may abstain from making a choice, valuable for preventing unsafe behaviors. Our contributions include:

1. Introducing **Self-REF**, a lightweight fine-tuning strategy that helps LLMs better learn when they should be confident in their predictions.

2. Introducing the **confidence-based routing** setting, where queries that a small local LLM is uncertain about can be optionally routed to a more powerful LLM.

3. Studying the **confidence-based rejection** setting, where the answer may be "none of the above" (e.g., if the LLM is not confident in any of the actions, it should defer to a fallback mechanism rather than choosing arbitrarily).

4. Demonstrating that Self-REF outperforms baselines in both the routing and rejection learning settings on four public datasets.

## 2. Related Work

In this section, we briefly introduce related studies on uncertainty quantification, LLM routing, and LLM rejection learning. More discussion and details about related work are provided in Appendix A.

### 2.1. Uncertainty Quantification in LLMs

Recent studies in LLM uncertainty have explored leveraging internal features like logits and attention scores to estimate uncertainty without additional training (Zhou et al., 2023; Mahaut et al., 2024; Turpin et al., 2024). Post-processing methods such as temperature scaling (Guo et al., 2017), prompting LLMs to verbalize confidence scores (Tian et al., 2023), iterative prompting (Abbasi Yadkori et al., 2024), and instruction learning/pretraining (Zhang et al., 2021; Neeman et al., 2023; Li et al., 2023; Liu et al., 2024; Cohen et al., 2024) have also shown improvements in calibration. In contrast to these methods, Self-REF allows the LLM to learn to estimate its confidence at fine-tuning time. Furthermore, this internal notion of confidence is aligned against correctness rather than the consistency of responses after re-sampling, as is studied in (Yona et al., 2024; Manakul et al., 2023; Lin et al., 2023). As noted in (Huang et al., 2023; Wang & Zhou, 2024; Chuang et al., 2025), uncertainty scores derived from LLMs may exhibit a weak correlation with actual prediction correctness, and the most consistent answers are often not the most correct. Finally, while much of the prior work focuses on calibration, our work goes beyond it and focuses on the downstream utility of confidence scores for LLM routing and rejection learning.

## 2.2. LLM Routing

In confidence-based query-routing scenarios, one line of research trains additional classifiers to route queries to LLMs based on observed performance metrics and routing data, forming routing pipelines with trained multi-agent systems or extra routers to boost performance (Ding et al., 2024; Ong et al., 2024; Stripelis et al., 2024; Jiang et al., 2024; Zhang et al., 2023). Another approach leverages LLM cascades, invoking increasingly complex LLMs depending on trained scoring functions which predict whether an answer is correct, and consistency of responses upon resampling (Chen et al., 2023b; Yue et al., 2023). Unlike prior work in LLM cascades, Self-REF trains the estimate of the probability of correctness into the LLM itself instead of an external model. Here, the estimates of confidence from Self-REF are aligned against the correctness of the predictions, rather than the consistency. Additionally, Self-REF computes the confidence in the autoregressive decoding step, allowing for LLM itself to condition on its generated output. This dynamic approach allows for more accurate and adaptive routing decisions based on real-time confidence assessments.

## 2.3. LLM Rejection Learning

Learning to reject (Cortes et al., 2016) was originally introduced to equip the Bayes classification model with a rejection option for predictions. Despite advancements in LLMs, their predictions can still be prone to uncertainty due to inherent knowledge limitations. This highlights the need for LLMs to have the ability to refuse to answer, especially in situations where it is infeasible to fallback upon more advanced models. LLMs can be trained for the rejection task through rejection knowledge injection (Chen et al., 2024; Zhang et al., 2024), an additional rejection LLM agent (Mao et al., 2023; 2024), and rejection-oriented loss adaptations (Li et al., 2024; Mohri et al., 2024). However, such approaches involving extra knowledge injection or loss adaptation may degrade the base performance of the LLM. In this work, we aim to equip LLMs with the capability for rejection learning based on their confidence levels, without the need for designing new loss functions, thereby preventing any potential degradation in the performance of downstream tasks.

# 3. Learning Confidence Tokens

## 3.1. Problem Setup and Notation

Consider any local large language model $\mathcal{M} : \mathcal{X} \rightarrow \mathcal{Y}$ with input query domain $\mathcal{X}$ and output prediction domain $\mathcal{Y}$. Here, *local* refers to a model that one has the ability to fine-tune (as opposed to API access). Let $\mathcal{D} = \{(x^{(i)}, y^{(i)})\}_{i=1}^{N}$ denote a dataset, where $x^{(i)}$ are queries and $y^{(i)}$ are ground truth answers to the queries. Let $\mathcal{D}_{\text{train}}$ denote the training

split of the dataset, $\mathcal{D}_{\text{val}}$ the validation split, and $\mathcal{D}_{\text{test}}$ the test split, where $\mathcal{D} = \mathcal{D}_{\text{train}} \cup \mathcal{D}_{\text{val}} \cup \mathcal{D}_{\text{test}}$. The goal in learning confidence tokens is to fine-tune $\mathcal{M}$ on $\mathcal{D}_{\text{train}}$ to use *confidence tokens*, two special tokens with trainable embeddings: as assessed on $\mathcal{D}_{\text{test}}$, the *confident token* <CN> should be generated when a model is confident that its prediction is correct (i.e., $\widehat{y}^{(i)} = y^{(i)}$), and the *unconfident token* <UN> should be generated when a model is not confident that its prediction is correct (i.e., $\widehat{y}^{(i)} \neq y^{(i)}$).

## 3.2. Self-REF

The Self-REF framework fine-tunes a base model $\mathcal{M}$ to use confidence tokens and also learn their embeddings. It involves *(i) confidence token annotation*, where data are augmented with confidence tokens, *(ii) Self-REF fine-tuning* on the augmented data, and *(iii) confidence score extraction*.

**Confidence Token Annotation** To incorporate model feedback into our training process, we first use the base model $\mathcal{M}$ to generate predictions $\widehat{y}^{(i)}$ for each sample $i = 1, \ldots, N_{\text{train}}$ in the dataset, where $\widehat{y}^{(i)} \in \mathcal{Y}$. For instances where $y^{(i)} \neq \widehat{y}^{(i)}$ (the prediction is incorrect), we construct an augmented dataset $\mathcal{D}'_{\text{train},\text{<UN>}} = \{(x^{(i)}, \widehat{y}^{(i)}\text{<UN>})\}$, where $\widehat{y}^{(i)}$<UN> denotes the concatenation of the incorrect prediction $\widehat{y}^{(i)}$ with an unconfident token <UN>. Conversely, for instances where the prediction is correct ($y^{(i)} = \widehat{y}^{(i)}$), we create the set $\mathcal{D}'_{\text{train},\text{<CN>}} = \{(x^{(i)}, \widehat{y}^{(i)}\text{<CN>})\}$, with $\widehat{y}^{(i)}$<CN> representing the correct prediction concatenated with a confident token <CN>. The augmented training dataset is then formed by combining these two datasets. This augmentation strategy enriches the training data by explicitly encoding the correctness of the model's predictions, thereby providing additional supervision that can guide the model to better distinguish between correct and incorrect responses. The detailed steps of confidence token annotation are shown in Algorithm 1.

**Self-REF Fine-tuning** Next, the base model $\mathcal{M}$ is fine-tuned using $\mathcal{D}'_{\text{train}}$. The confidence tokens <UN>and <CN>are special tokens during training, initialized as the average of other existing token embeddings. Gradients from the incorrect answers in the unconfident samples are masked out, to avoid increasing the probability of an incorrect answer given a query $P(\widehat{y}^{(i)} \neq y^{(i)} \mid x^{(i)})$. Instead, the model learns to increase the probabilities of confident tokens given correct answers $P(\text{<CN>} \mid \widehat{y}^{(i)} = y^{(i)}, x^{(i)})$, probabilities of unconfident tokens given incorrect answers $P(\text{<UN>} \mid \widehat{y}^{(i)} \neq y^{(i)}, x^{(i)})$, and probabilities of correct answers given the query $P(\widehat{y}^{(i)} = y^{(i)} \mid x^{(i)})$.

Note that the unconfident and confident tokens are generated end-to-end conditioned on both the prefix query and the answer that the LLM itself generated, in contrast to prior work (Ong et al., 2024; Ding et al., 2024; Stripelis et al.,

**Algorithm 1** Data Augmentation for Self-REF

---

**input** Base LLM $\mathcal{M}(\cdot)$ and original training data $\mathcal{D}_{\text{train}} = \{(x^{(i)}, y^{(i)})\}_{i=1}^{N_{\text{train}}}$.

**output** Augmented data $\mathcal{D}'_{\text{train}}$ with unconfident samples and confident samples. Use $\mathcal{M}(\cdot)$ to make predictions $\widehat{y}^{(i)}$ for the given input query $x^{(i)}$.

1: Create a set of unconfident samples:

$$\mathcal{D}'_{\text{train,<UN>}} = \{(x^{(i)}, \widehat{y}^{(i)}\text{<UN>}) : y^{(i)} \neq \widehat{y}^{(i)}\}_{i=1}^{N_{\text{train}}}.$$

2: Create a set of confident samples:

$$\mathcal{D}'_{\text{train,<CN>}} = \{(x^{(i)}, \widehat{y}^{(i)}\text{<CN>}) : y^{(i)} = \widehat{y}^{(i)}\}_{i=1}^{N_{\text{train}}}.$$

3: Mix the unconfident and confident data to form the combined augmented dataset, subsampling the unconfident samples with tunable proportion $\alpha \in [0, 1]$:

$$\mathcal{D}'_{\text{train}} = \text{subsample}_\alpha(\mathcal{D}'_{\text{train,<UN>}}) \cup \mathcal{D}'_{\text{train,<CN>}}.$$

---

2024) which typically routes based on the query alone with an extra router. By fine-tuning with the LLM's own predictions on the augmented training set, Self-REF customizes the notion of uncertainty as the indicator for routing and rejection to the LLM itself, rather than relying on the inherent uncertainty of a query.

**Confidence Score Extraction** After training, one can extract a continuous confidence score $c_{\mathcal{M}}(x^{(i)}, \widehat{y}^{(i)}) \in [0, 1]$ by getting the token probability of <CN>, normalized over the sum of the <UN> and <CN> token probabilities: $c_{\mathcal{M}}(x^{(i)}, \widehat{y}^{(i)}) = \frac{P(\text{<CN>})}{P(\text{<UN>})+P(\text{<CN>})}$. This continuous score gives us some control over navigating different tradeoffs in downstream settings.

### 3.3. Learning to Route and Reject

**Confidence-based Routing to a Larger Model** Often, constraints on computing resources and data may prevent researchers from fully fine-tuning very large LLMs. However, suppose one has the ability to fine-tune a smaller local LLM, as well as API access to a very large LLM. Given the limited capabilities of smaller LLMs and the high cost of very large LLMs, can we effectively route queries to the larger model based on the confidence level of the smaller model? How does one trade-off between cost and accuracy?

In confidence-based routing, answers where the smaller fine-tuned LLM is uncertain, i.e., $c_{\mathcal{M}}(x^{(i)}, \widehat{y}^{(i)}) < t$ for some chosen threshold $t \in [0, 1]$, will be routed to a more powerful LLM $\mathcal{M}_{\text{Large}}(\cdot)$. For answers with $c_{\mathcal{M}}(x^{(i)}, \widehat{y}^{(i)}) \geq t$ (i.e., certain), the answers from the local LLM $\mathcal{M}(\cdot)$ will be directly returned to the user.

**Confidence-based Rejection** A larger LLM may not be accessible in every situation. If one does not have access to a larger more capable model, one may still want to utilize measures of uncertainty in order to decide whether to rely on the model's prediction. This experimental setup studies the following scenario: "Sometimes, none of the provided options are good. How good am I at saying that I am not confident in any of the answers?"

To study this question, we create an evaluation set where half of the samples do not contain the ground truth, i.e., we remove all ground-truth information from $x^{(i)}$, and replace the label with "none of the above," $y^{(i)} = \emptyset$. Ideally, the model's confidence in any of the remaining answers should be low, and we can evaluate how good a policy for abstaining from answering the question is against this dataset. Thus, if $\mathcal{M}(\cdot)$ has low confidence, i.e., $c_{\mathcal{M}}(x^{(i)}, \widehat{y}^{(i)}) < t$ for a chosen threshold $t \in [0, 1]$, we treat it as if the model were to abstain from answering (i.e., $\widehat{y}^{(i)} = \emptyset$).

## 4. Experiment Setup

### 4.1. Datasets and Baselines

**Datasets** All experiments are conducted on the following four public datasets (more details in Appendix B):

- MMLU (Hendrycks et al., 2021a;b): The Massive Multitask Language Understanding (MMLU) dataset consists of multiple-choice questions covering a wide range of knowledge domains with 57 distinct tasks.

- OpenbookQA (Mihaylov et al., 2018): The OpenBookQA dataset is a question-answering dataset designed to test deep understanding through multi-step reasoning, common knowledge, and text comprehension, similar to open-book exams.

- GSM8K (Cobbe et al., 2021): GSM8K is a dataset containing graduate school math questions, where each question is collected from the Math World Problem Repository (Roy & Roth, 2015).

- MedQA (Jin et al., 2021): MedQA is a multiple-choice open domain question-answering dataset for problems collected from professional medical board exams.

**Baselines** We compare Self-REF to four state-of-the-art baselines for measuring model confidence, following the settings from existing work. Details about the prompts and hyper-parameters utilized in these baselines are given in Appendices B and C. The baselines are listed as follows:

- **Verbalizing Uncertainty** (Tian et al., 2023): Prompt the LLM with in-context learning to yield confidence scores

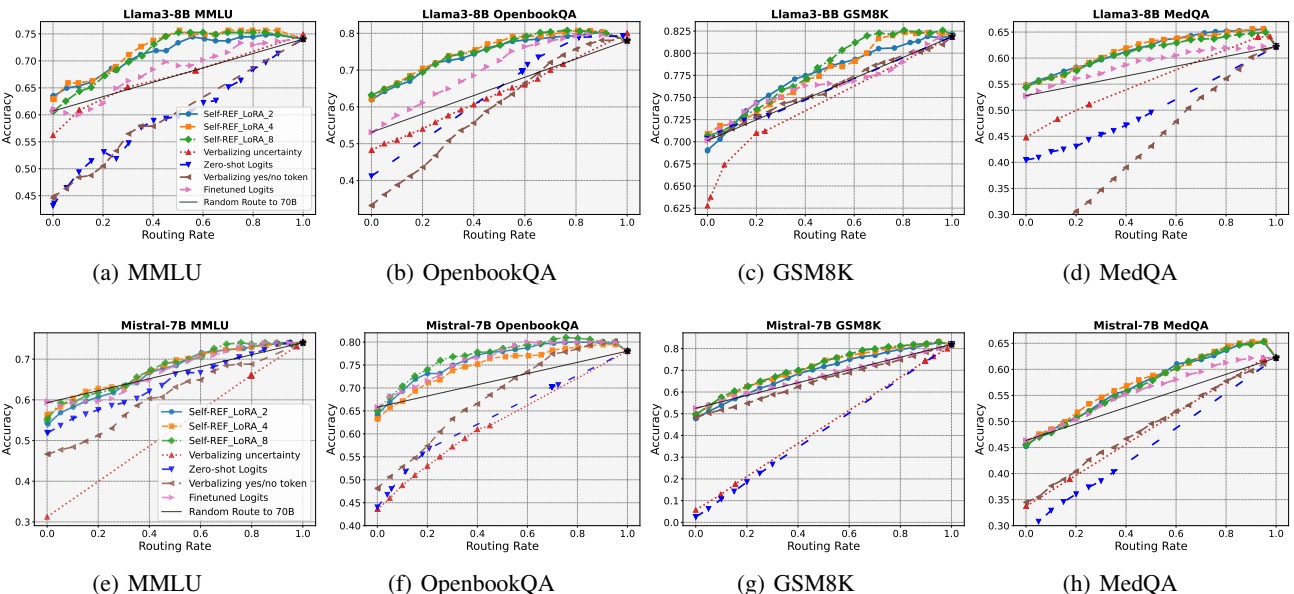

Figure 2: Overall accuracy vs. routing rate in confidence-based **routing** task. Routing is from local LLMs Llama3-8B-Instruct (row 1) and Mistral-7B-Instruct (row 2) to Llama3-70B-Instruct on MMLU (column 1), OpenbookQA (column 2), GSM8K (column 3), and MedQA (column 4). Routing rate of 0 gives pure local LLM performance, and routing rate of 1 gives pure Llama3-70B-Instruct performance. The solid black line corresponds to a baseline which chooses to route uniformly at random from the local fine-tuned LLM to the large LLM. Self-REF consistently outperforms baselines.

between 0 and 1. This encourages the LLM to generate its response while simultaneously estimating the confidence level of the response.

- **Verbalizing Yes/No Tokens** (Tian et al., 2023): Prompt the LLM with a question of whether the model is confident in its answers. The confidence score is calculated by normalizing the probabilities of the "Yes" and "No" tokens by their sum.

- **Zero-shot Logits** (Mahaut et al., 2024): Extract the probability of the generated prediction from the LLM without fine-tuning. In free-text generation questions, we average the probabilities of the tokens from the output answers. In the closed-form generation questions with single-token answers (e.g., multiple-choice questions), we directly select the probability of the predicted token.

- **Fine-tuned Logits** (Liu et al., 2024): Extract the probability of the generated prediction from a fine-tuned LLM. All methods for computing the probabilities of the outputs are identical to the "Zero-shot Logits" baseline. The confidence score is determined by the probabilities.

Taking these continuous-valued confidence scores from each baseline, one can apply thresholds and compare the utility of these scores versus those of Self-REF in the downstream routing and rejection learning settings.

## 4.2. Experiment Settings

**Confidence-based Routing Task** In the routing task, we aim to accurately route instances that the smaller LLMs (Llama3-8B-Instruct and Mistral-7B-Instruct) struggle with to the larger LLMs (Llama3-70B-Instruct) for improved performance. All experiments utilize Llama3-70B-Instruct with only its strong in-context learning capabilities during instance routing (no further fine-tuning). The routing decisions are determined by thresholding the normalized probability of the <CN> confidence token over the sum of the probabilities of <CN> and <UN> generated from Self-REF.

To characterize the trade-off between cost and performance, we analyze performance along several possible routing thresholds $t$. In particular, for a set of confidence scores $\mathcal{C}_{\mathcal{M},\mathcal{D}} = \{c_{\mathcal{M}}(x^{(i)}, \widehat{y}^{(i)})\}_{i=1}^{N}$, we set $t$ at quantiles, where $t = \mathcal{Q}_p(\mathcal{C}_{\mathcal{M},\mathcal{D}})$ for $p \in \{0, 1, 2, \ldots, 20\}$. Instances are routed to the larger LLM when $c_{\mathcal{M}}(x^{(i)}, \widehat{y}^{(i)}) < t$, so these thresholds correspond to routing approximately 0%, 5%, 10%, ..., 100% of total queries to the larger LLM.

**Confidence-based Rejection Learning Task** The rejection learning task is evaluated on the MMLU and OpenbookQA multiple-choice datasets. Self-REF does not explicitly train for the rejection learning task, as no additional samples are added where "None of the above" is a choice. This experiment tests whether the notion of confidence en-

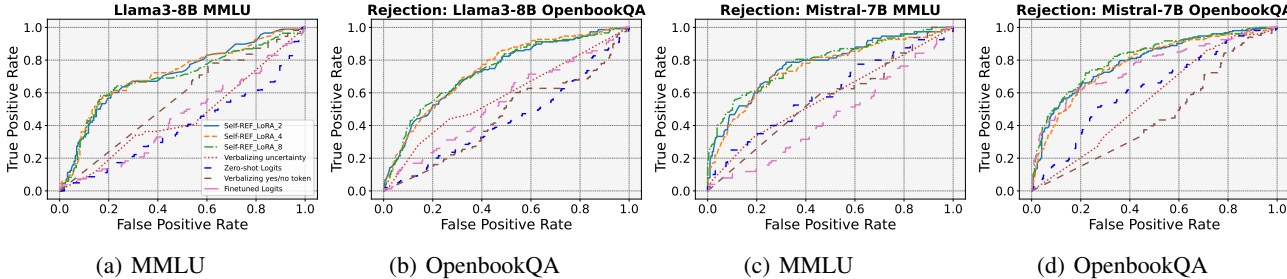

| (a) MMLU | (b) OpenbookQA | (c) MMLU | (d) OpenbookQA |

Figure 3: ROC curves of Llama3-8B-Instruct on MMLU (a) and OpenbookQA (b); and Mistral-7B-Instruct on MMLU (c) and OpenbookQA (d), for the **rejection learning** task. The true positive rate (recall) indicates the proportion of samples where "none of the above" is the ground truth that is correctly rejected. The false positive rate is the proportion of original samples that have been falsely rejected. Across all settings, Self-REF outperforms other baselines in learning to reject.

coded by <UN> and <CN> may nevertheless be useful for determining whether to reject all provided options.

To assess performance on the rejection learning task, we construct a specialized evaluation set where 50% of the time, the correct answer choice is removed from the choice list in the input question. Ideally, for any question where the ground truth choice was removed, the Self-REF fine-tuned LLM should append the <UN> token to any selected answer, since it would be incorrect. In order to examine the trade-off between rejecting too many samples vs. providing too many incorrect answers, we again utilize the $c_{\mathcal{M}}(x^{(i)}, \widehat{y}^{(i)})$ confidence score with several possible thresholds $t \in [0, 1]$.

**Implementation Details** We select Llama-3-8B-Instruct (Touvron et al., 2023) and Mistral-7B-v0.3-Instruct (Jiang et al., 2023) models as the pre-trained local base LLMs for Self-REF. The Llama3-70B-Instruct model (Touvron et al., 2023) is the larger, more powerful LLM to which unconfident queries are routed.

Parameter-efficient fine-tuning is performed using LoRA adapters with ranks 2, 4, and 8 following the observations from (Hayou et al., 2024). Under at most eight epochs of training, all other hyperparameters on model training are decided by grid search with the perplexity loss on validation sets. For more details about the hyper-parameters, see Appendix C.

## 5. Experiment Results

We conduct experiments to evaluate the performance of Self-REF, centered around the following three research questions: **RQ1:** Compared with state-of-the-art baselines, how does Self-REF perform on confidence-based routing? **RQ2:** How reliable are confidence scores from Self-REF for the rejection learning task? **RQ3:** How well-aligned are confidence-token-based scores of Self-REF and the actual probabilities of correctness?

### 5.1. Routing Performance (RQ1)

We analyze the routing performance from both a system-level accuracy and efficiency perspective.

**Overall Accuracy** Confidence-based routing using Self-REF consistently achieves the best accuracy vs. routing rate trade-off for all four datasets (MMLU, OpenbookQA, GSM8K, MedQA) and both local LLMs (Llama3-8B-Instruct model and Mistral-7B-Instruct) (Figure 2). Using Self-REF on Llama3-8B-Instruct for MMLU, confidence-based routing of just 39% of queries can achieve comparable performance to routing to the Llama3-70B-Instruct model alone. Similar observations hold for the OpenbookQA, GSM8K, and MedQA datasets, where routing the Llama3-8B-Instruct model with rates 49%, 65%, and 40%, respectively, can yield comparable performance to Llama3-70B-Instruct alone. For Self-REF on Mistral-7B-Instruct, the minimum routing rates to achieve parity with Llama3-70B-Instruct are 70%, 50%, 75%, and 70% for MMLU, Open-BookQA, GSM8K, and MedQA, respectively. Among the baselines, the fine-tuned logits tend to perform second-best.

**Efficiency Gains** Smaller local models often have lower latency (sec) during inference as well as lower monetary cost. Thus, routing only a fraction of queries to larger, more powerful models can significantly improve the efficiency of the overall system compared to relying entirely on the more powerful models. For each small LLM (Llama3-8B-Instruct and Mistral-7B-Instruct), we select the best routing rate as the minimum routing rate that achieves comparable performance to Llama3-70B-Instruct (see Figure 2 for the trade-off curves). Without sacrificing performance compared to Llama3-70B-Instruct, we observe that Self-REF achieves as much as a $2.03\times$ latency improvement for Llama3-8B-Instruct on MMLU and a $2.00\times$ latency improvement for Mistral-7B-Instruct on OpenbookQA (as shown in Table 1). This indicates the efficiency advantage of leveraging Self-REF in routing tasks.

Table 1: Per-token latency (sec) of smaller LLMs with routing using Self-REF vs. entirely using Llama3-70B-Instruct, while maintaining comparable overall accuracy (Acc.). Parentheses indicate the multiplicative speed-up.

| | MMLU | | OpenbookQA | | GSM8K | | MedQA | |
|---|---|---|---|---|---|---|---|---|
| | Acc. | Latency | Acc. | Latency | Acc. | Latency | Acc. | Latency |
| All in Llama3-70B-Instruct | 0.739 | 0.292 | 0.781 | 0.292 | 0.819 | 0.292 | 0.622 | 0.292 |
| Self-REF - Llama3-8B-Instruct | 0.739 | 0.145 (2.03×) | 0.781 | 0.172 (1.69×) | 0.816 | 0.220 (1.33×) | 0.619 | 0.147 (2.00×) |
| Self-REF - Mistral-7B-Instruct | 0.735 | 0.234 (1.25×) | 0.778 | 0.147 (2.00×) | 0.815 | 0.240 (1.20×) | 0.621 | 0.234 (1.25×) |

Table 2: Calibration metrics and routing rates of Self-REF and baselines using Llama3-8B-Instruct and Mistral-7B-Instruct, on the MMLU, OpenbookQA, GSM8K, and MedQA datasets. "Route" denotes the routing rate or proportion of queries that must be routed to achieve performance comparable to Llama3-70B-Instruct. The gray block refers to the best routing rate, "bold format" is the best calibration score, and "*" is the second best calibration score. Self-REF consistently achieves the best routing rate, but does not always achieve the best calibration score

| **Llama3-8B-Instruct** | MMLU | | | | OpenbookQA | | | | GSM8K | | | | MedQA | | | |
|---|---|---|---|---|---|---|---|---|---|---|---|---|---|---|---|---|
| | Route | ECE | BS | CE | Route | ECE | BS | CE | Route | ECE | BS | CE | Route | ECE | BS | CE |
| Verbalizing Uncertainty | 100% | 0.217 | 0.333 | 7.383 | 75% | 0.148 | 0.311 | 7.688 | 92% | 0.047 | 0.307 | 7.999 | 100% | 0.069 | 0.331 | 1.206 |
| Verbalizing Yes/No Token | 100% | 0.466 | 0.397 | 10.281 | 95% | 0.291 | 0.415 | 5.319 | 100% | 0.107 | 0.384 | 8.369 | 100% | 0.090 | 0.470 | 11.276 |
| Zero-shot Logits | 100% | 0.347 | 0.418 | 1.733* | 90% | 0.225 | 0.496 | 1.705 | 90% | 0.027 | 0.292* | 3.312 | 100% | 0.108 | 0.736 | 3.767 |
| Finetuned Logits | 90% | 0.081 | **0.206** | **0.602** | 70% | 0.095 | **0.238** | **0.676** | 70% | 0.055 | **0.256** | **0.930** | 80% | **0.059** | **0.265** | **0.844** |
| Self-REF - Llama3-8B-Instruct | 39% | **0.040** | 0.320* | 11.534 | 49% | **0.090** | 0.254* | 1.118* | 65% | **0.019** | 0.400 | 3.088* | 40% | 0.066* | 0.278* | 1.085* |

| **Mistral-7B-Instruct** | MMLU | | | | OpenbookQA | | | | GSM8K | | | | MedQA | | | |
|---|---|---|---|---|---|---|---|---|---|---|---|---|---|---|---|---|
| | Route | ECE | BS | CE | Route | ECE | BS | CE | Route | ECE | BS | CE | Route | ECE | BS | CE |
| Verbalizing Uncertainty | 100% | 0.126 | 0.313 | 8.725 | 100% | 0.221 | 0.434 | 10.846 | 100% | 0.163 | 0.669 | 3.822 | 100% | 0.087 | 0.564 | 2.480 |
| Verbalizing Yes/No Token | 100% | 0.297 | 0.345 | 4.380 | 100% | 0.252 | 0.394 | 5.863 | 100% | 0.146 | 0.821 | 23.98 | 100% | 0.090 | 0.579 | 17.14 |
| Zero-shot Logits | 95% | 0.311* | 0.353 | 1.643 | 100% | 0.232 | 0.442 | 2.498 | 100% | **0.046** | 0.398 | 1.957 | 100% | 0.087 | 0.507* | 2.409 |
| Finetuned Logits | 95% | **0.173** | **0.263** | **0.751** | 50% | **0.143** | 0.207* | **0.627** | 100% | 0.050* | 0.323 | 1.236* | 90% | **0.023** | 0.226 | **0.633** |
| Self-REF - Mistral-7B-Instruct | 70% | 0.369 | 0.293* | 1.021* | 40% | 0.157* | **0.193** | 0.869* | 75% | 0.068 | **0.273** | **0.968** | 70% | 0.037* | **0.226** | 0.653* |

## 5.2. Rejection Learning Performance (RQ2)

In this section, we analyze the utility of confidence scores from Self-REF for the purpose of learning to abstain from answering. Without incorporating additional knowledge of the rejection task during training, Self-REF and baselines rely solely on thresholding the confidence scores to determine whether to reject a prediction. In safety-critical applications where one may hope to detect cases that should be abstained from answering (i.e., high recall on the reject option) while retaining performance on the task at hand (i.e., low false positive rate), it is useful to examine the ROC curve, which makes this trade-off. Across all thresholds, we observe that on both MMLU and OpenbookQA dataset, Self-REF outperforms all baselines on the rejection learning task (Figure 3). Because the confidence tokens are trained and conditioned on the correctness of the predicted answers, Self-REF has a higher likelihood of being aware of the correctness of its generated responses using these tokens. In this manner, Self-REF may better distinguish when there is no correct answer to choose, resulting in better rejection learning capabilities.

## 5.3. Calibration with Utility (RQ3)

Here, we analyze the correlation between calibration, or how well predicted probabilities align with actual probabilities, and utility for downstream routing. Following the settings from Tian et al. (2023); Błasiok et al. (2023), three calibration metrics are used to assess the calibration: (1) expected calibration error, ECE (Guo et al., 2017); (2) Brier Score, BS (Brier, 1950); and (3) the cross-entropy score, CE. Alongside these calibration metrics, we also show the least rates of routing required for each technique to attain comparable performance to Llama3-70B-Instruct. As shown in Table 2, although Self-REF often achieves the best or second-best calibration scores, better-calibrated methods do not guarantee optimal results for routing. This aligns with observations from prior work (Huang et al., 2023), which found that well-calibrated confidence scores do not necessarily imply a strong correlation with the correctness of the predictions. Overall, Self-REF achieves superior routing performance across all datasets and local LLMs, even though it only outperforms baseline methods under most of the calibration assessments.

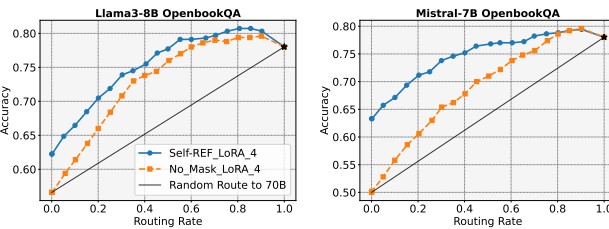

Figure 4: Impact of gradient masking during Self-REF fine-tuning versus not (No Mask), assessed on the routing task.

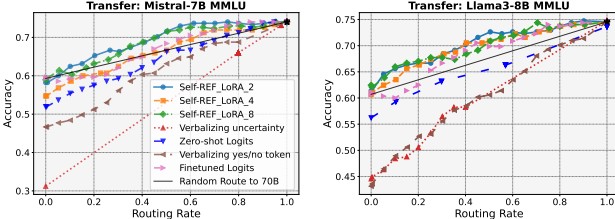

Figure 5: Transferability of Self-REF, assessed on the routing task. Local LLMs are fine-tuned with Self-REF on OpenbookQA and inference is run on the MMLU dataset.

### 5.4. Ablation Study on Gradient Masking

To evaluate the impact of gradient masking on downstream fine-tuning performance, we consider Self-REF with and without gradient masking on the OpenbookQA dataset. For both Llama3-8B-Instruct and Mistral-7B-Instruct, we find that applying gradient masking consistently improves routing task accuracy (Figure 4). This supports the intuition behind gradient masking: by masking gradients on samples with <UN> tokens, we prevent the model from learning spurious patterns based on uncertain inputs.

### 5.5. Transferability of Confidence Token

We further explore the transferability of Self-REF across unseen datasets (i.e., not included in the training data). With fine-tuned LoRA weights on OpenbookQA, we test on MMLU for Llama3-8B-Instruct and Mistral-7B-Instruct. Self-REF outperforms the baselines with good routing performance on the MMLU dataset even when using the transferred LoRA weights of the OpenbookQA dataset (Figure 5). To achieve parity with the 70B model, we observe that routing rates of 75% and 70% are required on MMLU under Mistral-7B-Instruct and Llama3-8B-Instruct, respectively. Furthermore, compared to the direct fine-tuning on MMLU (i.e., a 70% routing rate on Mistral-7B-Instruct and a 40% routing rate on Llama3-8B-Instruct, Figure 2), transferred Self-REF achieves competitive routing performance with only a slight degradation in routing rates. This demonstrates the potential of Self-REF to be trained for general, multi-task purposes due to its transferability.

---

**Question-1:** What is the difference between a male and a female catheter?

**Subject:** Clinical Knowledge
**Choices:** [A] Male and female catheters are different colors. [B] Male catheters are longer than female catheters. [C] Male catheters are bigger than female catheters. [D] Female catheters are longer than male catheters
**Ground Truth Answer**: **[B]** Male catheters are longer than female catheters.
**Self-REF-8B:** (✗) **[D]** Female catheters are longer than male catheters. **<UN> → Route**
**Larger LLM-70B:** (✓) **[B]** Male catheters are longer than female catheters.

---

**Question-2:** A certain pipelined RISC machine has 8 general-purpose registers R0, R1, . . . , R7 and supports the following operations. [Truncated 50 words]. If the contents of these three registers must not be modified, what is the minimum number of clock cycles required for an operation sequence that computes the value of AB + ABC + BC?

**Subject:** College Computer Science
**Choices:** [A] 5 [B] 6 [C] 7 [D] 8
**Ground Truth Answer**: **[B]** 6.
**Self-REF-8B:** (✗) **[C]** 7. **<CN> → Overconfident**
**Larger LLM-70B:** (✓) **[B]** 6.

---

**Question-3:** This question refers to the following information. Albeit the king's Majesty justly and rightfully is and ought to be the supreme head of the Church of England, [Truncated 230 words]; From the passage, one may infer that the English Parliament wished to argue that the Act of Supremacy would

**Subject:** High School European History
**Choices:** [A] give the English king a new position of authority [B] give the position of head of the Church of England to Henry VIII alone and exclude his heirs [C] establish Calvinism as the one true theology in England [D] end various forms of corruption plaguing the Church in England
**Ground Truth Answer**: **[D]** end various forms of corruption plaguing the Church in England
**Self-REF-8B:** (✓) **[D]** end various forms of corruption plaguing the Church in England. **<UN> → Underconfident**
**Larger LLM-70B:** (✓) **[D]** end various forms of corruption plaguing the Church in England.

---

Figure 6: Examples of routing from Llama-8B-Instruct to Llama3-70B-Instruct, on the MMLU dataset.

### 5.6. Case Studies

While Self-REF significantly outperforms other methods in routing accuracy, some failure cases still occur, particularly on niche subjects or tasks requiring nuanced reasoning. In underconfident cases, the model often hesitates on context-heavy (i.e., High School Europe History in Figure 6) or ambiguous subjects, highlighting its sensitivity to topics requiring broader reasoning. Overconfident failure cases occur in technical or highly structured domains (i.e., College Computer Science in Figure 6), where the model appears

familiar but fails to grasp nuanced or edge-case reasoning. This phenomenon is also observed in the jurisprudence domain, where subtle distinctions in legal language and context-specific reasoning often lead to overconfident yet incorrect predictions. Conversely, correctly predicted confident cases typically involve broad conceptual topics, due to training coverage and clearer correctness signals. Meanwhile, incorrect but uncertain cases often correspond to detail-intensive subjects, where the model appropriately expresses hesitation due to limited recall or understanding. For more details see Appendix E.

## 6. Discussion

Self-REF delivers comparable or even superior system-level routing performance compared to the non-finetuned 70B model. Note that a non-finetuned 70B model is used to mimic the common setting where practitioners may not have sufficient resources to fine-tune such a large model, but can instead fine-tune a smaller model on the task of interest. In this section, we dive into the following questions: *(1) Why can Self-REF after confidence-based routing achieve better performance than the non-finetuned 70B model?* and *(2) How can one navigate the tradeoffs in routing and rejection?*

For *question (1)*, we believe that one possible reason behind this phenomenon is that fine-tuning with Self-REF does not degrade the model's performance and effectively introduces correct knowledge, such as niche astronomy knowledge or complicated moral disputes, that the model does not inherently possess. Additionally, since the error distributions of two independent models are unlikely to overlap perfectly, the ability to route can serve as a form of ensembling of models. In Appendix D, we provide some case studies from MMLU that the powerful Llama3-70B-Instruct model answered incorrectly, but Self-REF with the smaller Llama-8B model answered confidently and correctly. As shown in Appendix Figure 8, the Llama3-70B-Instruct may lack specific knowledge to judge a question of complicated moral disputes without fine-tuning and injecting the knowledge directly. More case studies are provided in Appendix E.

For *question (2)*, Self-REF provides a continuous-valued confidence score which enables downstream decision-makers a granular level of control. By contrast, we observe that certain methods for extracting confidence scores, such as verbalizing uncertainty, can suffer from mode collapse, where the scores cluster around a limited set of numbers (Figure 2, red dotted triangle lines). To appropriately leverage the granular control provided by Self-REF, one must consider the parameters of the downstream setting.

When navigating the accuracy vs. routing rate trade-off, one must consider the setting's cost of routing. For example, if one is utilizing an LLM on a rover on Mars, then the latency

and monetary cost of routing to a larger LLM on a server on Earth is enormous, and one may want to avoid routing at all costs. If one has a limited budget for API calls to a more powerful LLM, then one may also want to route sparingly. On the other hand, if monetary cost is not an issue, and latency is not critical, then one can have the entire trade-off curve available to them, and simply choose the point that obtains the highest accuracy.

When navigating the rejection learning setting, one must consider the risk of an incorrect answer. For example, if one is using an LLM to inform medical treatment, then one may choose to abstain from answering more often. On the other hand if one is using LLM for creative endeavors and correctness is not vital, then one may choose to let the LLM make its predictions and rarely abstain. Overall, these settings are highly customizable and controllable depending on the parameters that one is faced with in reality.

Through confidence-based routing and rejection, Self-REF enhances the overall performance and safety of a system while reducing cost. In our analysis, we measure cost primarily in terms of GPU hours and latency, excluding network transmission time between models which could be hosted on different machines (which is subject to varying signal quality, setup costs, and network congestion). Future research in downstream applications could seek to better characterize these networking costs, which may contribute substantially to savings gained from confidence-based routing.

## 7. Conclusion

In this work, we introduce Self-REF, a lightweight fine-tuning strategy that incorporates confidence tokens, enabling LLMs to better express their confidence levels in a way that closely correlates with prediction correctness. These confidence scores are valuable for confidence-based routing and rejection learning, which can improve system performance, efficiency, and output reliability for both LLM routing and LLM rejection learning purposes. Our findings suggest that integrating confidence tokens into LLMs is a promising step toward improving their reliability and safety, particularly in real-world applications where errors carry significant costs.

More broadly, Self-REF opens new avenues for research into confidence-based routing and rejection. For instance, Self-REF could be extended to route queries to multiple specialized LLMs rather than a single powerful model. Another direction involves finer-grained annotation schemes, where individual sentences are labeled as confident or uncertain. Based on confidence, the system could route to different models, stop early, or re-invoke the reasoning processes to refine outputs. For confidence-based rejection, one could expand Self-REF to incorporate explanations of rejections or to suggest actionable clarifying information.

## Impact Statement

This paper presents work whose goal is to advance the field of Machine Learning. There are many potential societal consequences of our work, none of which we feel must be specifically highlighted here.

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

# Appendix

# A. Related Work

## A.1. Uncertainty Quantification in LLMs

There is a growing body of recent works examining the calibration of LLMs. One line of research (Zhou et al., 2023; Mahaut et al., 2024; Turpin et al., 2024) leverages the internal features of LLMs (e.g., logits, attention scores) to assess and estimate uncertainty without requiring additional training or fine-tuning. Improvements in calibration have also been observed from post-processing approaches such as temperature scaling or Platt scaling (Guo et al., 2017), as well as prompting approaches asking the LLM to verbalize its predicted confidence scores (Tian et al., 2023). Another line of work (Zhang et al., 2021; Neeman et al., 2023; Li et al., 2023; Liu et al., 2024) focuses on RLHF or instruction learning for achieving better prediction calibration.

In addition to model-centric analysis, another school of approaches (Hou et al., 2024; Kuhn et al., 2023; Duan et al., 2024) focuses on examining the semantics and ambiguity in the given input questions to estimate uncertainty in LLMs. On the other hand, (Huang et al., 2023) have found that uncertainty scores derived from logits and verbalized scores exhibit a weak correlation with the correctness of LLM predictions across several models and datasets. Thus, while calibration may be a desirable characteristic of uncertainty estimates, we note that this work goes beyond calibration as a goal, focusing more on the utility of model confidence scores downstream, for tasks such as instance routing and rejection learning.

## A.2. LLM Routing

LLM routing, where all or part of a query is directed from one LLMs to different LLMs, has been studied in the literature in a few capacities: (1) routing entire queries to different LLMs, (2) retrieval-augmented generation (RAG) systems that route part of the query to an external retrieval system, and (3) speculative decoding which utilizes outputs from models of varying complexity in the decoding step.

(1) In the query-routing scenario, one line of work trains additional classification models to route queries to other LLMs for answers based on designated performance metrics (Ding et al., 2024; Hu et al., 2024; Huang et al., 2024) and routing data (Ong et al., 2024; Stripelis et al., 2024), which eventually form a routing pipeline with multi-agent systems (Jordan & Jacobs, 1994; Jiang et al., 2024) to boost the performance. Another line of work leverages routers to measure utility-efficiency trade-offs, aiming to reconstruct model architectures to reduce inference overhead in terms of cost and execution time while maintaining utility (Chen et al., 2023b; Yue et al., 2023). (2) In RAG systems, routing provides effective and efficient navigation to specific data sources, offering efficient and effective indicators for RAG systems to retrieve. A notable approach is Self-RAG (Asai et al., 2024), which fine-tunes an LLM retriever using augmented data with predefined routing indicators to optimize retrieval performance. This approach improves RAG systems by enabling more efficient retrieval through dynamic routing decisions of each given sentence in the corpus based on predicted routing indicators generated by the fine-tuned LLM retriever. (3) In speculative decoding, the entire decoding process is carried out through an iterative collaboration between a smaller LLM and a larger LLM. Each final output token sequence can be generated by either a smaller LLM or a larger LLM. Specifically, the token decoding process will route to a larger language model when the smaller model either declines to generate or encounters a failure in token generation (Chen et al., 2023a; Leviathan et al., 2023). Different from the routing concepts in the work of training extra query routers, we route the queries based on the model confidence of each LLM output answer. The confidence estimations in our work are predicted given the output answers during the autoregressive decoding process, which makes the model confidence more aligned with the correctness output answer.

## A.3. LLM Rejection Learning

Learning to reject (Cortes et al., 2016) has been initially proposed to offer the Bayes classification model a rejection option as the final prediction. However, with the rapid advancement of LLMs, the issue of uncertainty in LLMs' predictions may stem from their limited inherent knowledge, necessitating an option or mechanism for LLMs to refuse to answer, especially when no advanced LLMs are helpers for two-step verification or answering in action. The rejection decisions made by LLMs can enhance the reliability of their generated predictions. One group of work offers LLMs a rejection option while they generate the responses by finetuning or instruction learning (Chen et al., 2024) with newly designed rejection loss. Another group of work trains the additional rejection LLM agents to issue a rejection instruction after the predictor LLMs have made their prediction (Kamath et al., 2020; Mozannar & Sontag, 2020; Mao et al., 2023; 2024). Unlike the training paradigm of

traditional machine learning models, establishing a new rejection loss with new knowledge in fine-tuning LLMs may result in sub-optimal performance and potentially degrade the abilities of pre-trained LLMs in next-token prediction. A more effective approach is to adjust LLMs using the original training data and loss functions (Li et al., 2024; Mohri et al., 2024) while incorporating the capability for rejection learning without altering the foundational training dynamics. Moreover, training additional agents to handle rejection learning can introduce significant latency and may be overfitted, making the approach impractical for real-world applications. In this work, we aim to equip predictive LLMs with the capability for rejection learning by enabling them to reveal their confidence levels, without the need for designing new loss functions, thereby preventing any potential degradation in the performance of downstream tasks.

## B. Details about Baseline and Datasets

### B.1. Verbalizing Confidence Prompt Usage

We provide a listing of the evaluation prompts in Table 3 utilized in assessing the performance of the baselines among all four datasets. The first sections reveal the evaluation prompt for "Verbalizing Uncertainty;" the second sections demonstrate the evaluation prompt for "Verbalizing Yes/No Tokens;" and third batches demonstrate the evaluation prompt for "Zero-shot/Fine-tuned Logits."

| Confidence Baselines | Dataset | Evaluation Prompts |
| --- | --- | --- |
| Verbalizing Uncertainty | MMLU | Answer the input question. For each of the answer choices A, B, C, and D. Output your confidence that it is the correct answer. The total values must equal ONE. <Task Description>. You must follow the format for your final answer: '##Answer:<Your Answer> ##Confidence:<Your confidence>' |
| Verbalizing Uncertainty | OpenbookQA MedQA | Answer the input question. For each of the answer choices A, B, C, and D. Output your confidence that it is the correct answer. The total values must equal ONE. You must follow the format for your final answer: '##Answer:<Your Answer> ##Confidence:<Your confidence>' |
| Verbalizing Uncertainty | GSM8K | Output your confidence that it is the correct answer. The total values must equal ONE. You must follow the format for your final answer: '##Answer:<Your Answer> ##Confidence:<Your confidence>'. Let's think step by step and output the answers and confidence score following the format only." |
| Verbalizing Yes/No Tokens | MMLU | Please answer the questions first. After providing your answers, indicate your confidence level in your responses. <Task Description>. Respond with 'Yes' if you are confident and 'No' if you are not confident. |
| Verbalizing Yes/No Tokens | OpenbookQA MedQA | Please answer the questions first. After providing your answers, indicate your confidence level in your responses. Respond with 'Yes' if you are confident and 'No' if you are not confident. |
| Verbalizing Yes/No Tokens | GSM8K | Please answer the questions first. After providing your answers, indicate your confidence level in your responses. Respond with 'Yes' if you are confident and 'No' if you are not confident. You must follow the format for your final answer: '##Answer:<Your Answer> ##Confidence:<Your confidence>'. Let's think step by step and output the answers and confidence score following the format only." |
| Zero-shot Logits Fine-tuned Logits | MMLU OpenbookQA MedQA | Answer the question by choosing one of the answer choices A, B, C, or D. |
| Zero-shot Logits Fine-tuned Logits | GSM8K | Let's think step by step. |

Table 3: Evaluation prompts for verbalizing uncertainty on local LLMs under different dataset.

## B.2. Finetuned Logits Training

The hyperparameter settings (shown in Table 5) used for LoRA training of the "Finetuned Logits" baseline are as follows. The LoRA dimension are set from either 2, 4, and 8, and the reported performance of "Finetuned Logits" is the one with highest downstream task accuracy.

Table 4: Hyper-parameter settings in "Fine-tuned Logits" baseline.

|  | Dataset | MMLU | OpenbookQA | GSM8K | MedQA |
|---|---|---|---|---|---|
| Llama3-8B-Instruct | Optimizer | Adam | Adam | Adam | Adam |
|  | Warm Start | 200 | 200 | 200 | 2000 |
|  | Total Training Data # | 9,985 | 4,957 | 7,473 | 10,000 |
| Mistral-7B-Instruct | Optimizer | Adam | Adam | Adam | Adam |
|  | Warm Start | 200 | 200 | 200 | 200 |
|  | Total Training Data # | 9,985 | 4,957 | 7,473 | 10,000 |

## B.3. Details of Datasets

The experiment are conducted on four different open-sourced datasets. The details of the datasets are provided as follows:

- **MMLU (Hendrycks et al., 2021b;a).** A diverse benchmark covering 57 subjects across humanities, social sciences, STEM, and professional domains. It evaluates a model's ability to perform zero-shot multiple-choice question answering using knowledge acquired during pretraining. In the experiment, we evaluate the model on val set.

- **OpenbookQA (Mihaylov et al., 2018).** A 5,957 multiple-choice question answering dataset focused on elementary science. It requires combining a small "open book" of core science facts with broader common knowledge and reasoning.

- **GSM8K (Cobbe et al., 2021).** A dataset of grade-school level math word problems designed to test arithmetic reasoning. It includes both questions and detailed step-by-step solutions, making it ideal for evaluating models' reasoning and intermediate computation skills. We assess the model on 1,331 test set.

- **MedQA (Jin et al., 2021).** A large-scale multiple-choice medical question answering dataset based on real-world medical licensing exams with 12,723 questions. It challenges models with clinically relevant questions requiring specialized medical knowledge and multi-step reasoning.

## C. Experiment and Model Training Details

### C.1. Hyper-parameter Settings

Our experiments on the dataset are conducted under Self-REF. We focus on the confidence-based routing and rejection learning tasks, following the pipeline of **Base Model Training** and **Confidence Token Inference**. The details of each step is shown as follows:

**Base Model Training**    In this work, Self-REF is fine-tuned on local LLMs (i.e., Llama3-8B-Instruct and Mistral-7B-Instruct) under the following hyper-parameters. We apply LoRA adapters to every query, key, and value (Q-K-V) layers, the token embedding layers, and the final linear layers in the local LLMs with the batch size of 4 and learning rate of 1e-4. Fixing the overall dataset size, the parameter $\alpha$ is tuned based on performance on the validation set, selecting from a ratio of unconfident to confident data of 1:1, 1:2, 1:3, 1:4, and 1:5. On one hand, including more confident samples helped fine-tune the model to perform better on the downstream task; on the other hand, a sufficient number of unconfident samples was necessary to teach the model when to express uncertainty. For the ratio of training dataset, we only adjust the OpenbookQA and MedQA dataset in Mistral-7B-Instruct model to prevent the overfitting situation. We retain the original dataset settings with the correctness and incorrectness directly inferred by local LLMs.

|  | Dataset | MMLU | OpenbookQA | GSM8K | MedQA |
|---|---|---|---|---|---|
| Llama3 | LoRA Dimension | 2, 4, 8 | 2, 4, 8 | 2, 4, 8 | 2, 4, 8 |
|  | Optimizer | Adam | Adam | Adam | Adam |
|  | Weight Decay | $10^{-2} \sim 10^{-1}$ | $10^{-5} \sim 10^{-1}$ | $5 \times 10^{-1} \sim 10^{-2}$ | $10^{-2} \sim 7 \times 10^{-1}$ |
|  | Warm Start | 200 | 200 | 200 | 200 |
|  | Ratio <CN>*: <UN> | 2.79 | 1.01 | 4.47 | 2.74 |
|  | Total Training Data | 9,985 | 4,957 | 7,473 | 10,000 |
|  | Training Time (Hours) | $\sim$4 | $\sim$1 | $\sim$2.5 | $\sim$4 |
| Mistral | LoRA Dimension | 2, 4, 8 | 2, 4, 8 | 2, 4, 8 | 2, 4, 8 |
|  | Optimizer | Adam | Adam | Adam | Adam |
|  | Weight Decay | $10^{-2} \sim 5 \times 10^{-1}$ | $9 \times 10^{-2} \sim 10^{-1}$ | $5 \times 10^{-2} \sim 10^{-1}$ | $10^{-2} \sim 7 \times 10^{-1}$ |
|  | Warm Start | 200 | 200 | 200 | 200 |
|  | Ratio <CN>*: <UN> | 1.37 | _2.25_ | 1.01 | _2.05_ |
|  | Total Training Data | 9,985 | 3,395 | 7,473 | 7,557 |
|  | Training Time (Hours) | $\sim$4 | $\sim$0.5 | $\sim$2 | $\sim$4 |

Table 5: Hyper-parameters and model structures settings in Self-REF. <CN>* denotes the amount of <CN> data, which depends on the number of questions that local LLMs answer correctly. The underlined ratios indicate the proportion of queries with <UN> that are downsampled to prevent overfitting.

**Self-REF Inference**    During the inference time, the temperature is set as 0, top-p as 1.0, and all decoding processes are greedy search. All other sampling strategies are forbidden with the fixed random seed 2024 for the reproducibility.

### C.2. Computation Infrastructure

The experiments are conducted based on the following physical computing infrastructure in Table 6. This setup provided the necessary computational resources to efficiently train and evaluate our models.

| Device Attribute | Value |
|---|---|
| Computing infrastructure | GPU |
| GPU model | Nvidia-A100 |
| GPU number | 8 |
| GPU Memory | 80G |

Table 6: Computing infrastructure for the experiments.

# D. Case studies of Self-REF Routing

We now perform a case study of Self-REF on the MMLU dataset. Recall from the results that Self-REF can effectively learn confidence tokens without degrading performance. We here showcase two types of examples: (1) cases that Self-REF finetuning on Llama3-8B-Instruct got correct but Llama3-70B-Instruct got incorrect (Figure 6), and (2) cases that Llama3-8B-Instruct initially got incorrect but were corrected by routing to Llama3-70B-Instruct using Self-REF confidence scores (Figure 6).

---

**Question-1:** Baron admits that the versions of the ticking bomb hypothetical she discusses are "stunningly stupid," but she claims this is actually evidence of?

**Subject:** Moral disputes

**Choices:** [A] the stupidity of most traditional philosophical examples. [B] a general lack of intelligence among people with advanced degrees. [C] the wrongness of torture. [D] the readiness on the part of many intelligent people to see torture as the best solution to deal with terrorism.

**Groun Truth Answer**: **[D]** the readiness on the part of many intelligent people to see torture as the best solution to deal with terrorism.

**8B Response:** (✓) **[D]** the readiness on the part of many intelligent people to see torture as the best solution to deal with terrorism. **<CN>**

**70B Response:** (✗) **[A]** the stupidity of most traditional philosophical examples.

---

**Question-2:** Functions of the law include all but which of the following?

**Subject:** Jurisprudence

**Choices:** [A] maximizing individual freedom. [B] providing a basis for compromise. [C] keeping the peace. [D] promoting the principles of the free enterprise system.

**Ground Truth Answer**: **[D]** promoting the principles of the free enterprise system.

**8B Response:** (✓) **[D]** the readiness on the part of many intelligent people to see torture as the best solution to deal with terrorism. **<CN>**

**70B Response:** (✗) **[A]** maximizing individual freedom.

---

**Question-3:** Question-263: What is the Second Gem in Buddhism??

**Subject:** World Religions

**Choices:** [A] The Dharma [B] The Sangha [C] The Buddha [D] The Bodhisattva

**Ground Truth Answer**: **[A]** promoting the principles of the free enterprise system.

**8B Response:** (✓) **[A]** The Dharma. **<CN>**

**70B Response:** (✗) **[B]** The Sangha.

---

Figure 7: The examples on MMLU dataset, where the Llama-8B model correctly answers a query after routing, while the non-fine-tuned Llama3-70B-Instruct model answers incorrectly.

---

**Question-1:** Statement 1| Every function from a finite set onto itself must be one to one. Statement 2 | Every subgroup of an abelian group is abelian.

**Subject:** Abstract Algebra

**Choices:** [A] True, True [B] False, False [C] True, False [D] False, True

**Ground Truth Answer**: **[A]** True, True.

**8B Response:** (✗) **[C]** True, False. **<UN> → Route**

**70B Response:** (✓) **[A]** True, True.

---

**Question-2:** A model airplane flies slower when flying into the wind and faster with wind at its back. When launched at right angles to the wind a cross wind its groundspeed compared with flying in still air is?

**Subject:** Conceptual Physics

**Choices:** [A] the same [B] greater [C] less [D] either greater or less depending on wind speed

**Ground Truth Answer**: **[B]** greater.

**8B Response:** (✗) **[D]** either greater or less depending on wind speed. **<UN> → Route**

**70B Response:** (✓) **[B]** greater.

---

**Question-3:** Two long parallel conductors carry 100 A. If the conductors are separated by 20 mm, the force per meter of length of each conductor will be?

**Subject:** Electrical Engineering

**Choices:** [A] 100 N. [B] 0.1 N. [C] 1 N. [D] 0.01 N.

**Ground Truth Answer**: **[B]** 0.1 N.

**8B Response:** (✗) **[C]** 1 N. **<UN> → Route**

**70B Response:** (✓) **[B]** 0.1 N.

---

Figure 8: The routing examples with <UN> on MMLU dataset, where the Llama-8B model incorrectly answers a query initially. After routing to Llama3-70B-Instruct, the queries answer correctly.

# E. Case studies by Subject Area

To better understand the model's calibration behavior, we analyze the top five subject categories across four groups based on prediction correctness and confidence levels:

- **Correct Predictions + <UN> (Underconfident)**:

  – Computer Security
  – High School Biology
  – High School European History
  – Human Sexuality
  – Miscellaneous

- **Incorrect Predictions + <CN> (Overconfident)**:

  – College Computer Science
  – Conceptual Physics
  – High School Computer Science
  – High School Microeconomics
  – Jurisprudence

- **Correct Predictions + <CN> (Confident and Accurate)**:

  – International Law
  – College Biology
  – Moral Disputes
  – Philosophy
  – U.S. Foreign Policy

- **Incorrect Predictions + <UN> (Unconfident and Incorrect)**:

  – Abstract Algebra
  – Anatomy
  – College Chemistry
  – College Medicine
  – Econometrics

This categorization reveals several noteworthy trends:

- **Correct + Underconfident:** The model tends to be underconfident on subjects that involve ambiguity, contextual reasoning, or non-standard formats. For example, topics like human sexuality and European history often require nuanced interpretation. The presence of "Miscellaneous" suggests increased uncertainty when the question content falls outside well-defined training distributions.

- **Incorrect + Overconfident:** Overconfidence is particularly common in structured, technical domains such as computer science and physics. This behavior likely arises from overfitting to frequent patterns seen in pretraining, while failing on nuanced reasoning or edge cases (e.g., legal intricacies in jurisprudence).

- **Correct + Confident:** In domains such as philosophy and international law, the model demonstrates reliable performance and appropriate confidence. These topics often involve conceptual reasoning with clear evaluative criteria, which may contribute to better calibration.

- **Incorrect + Underconfident:** In contrast, the model appropriately exhibits low confidence in subjects requiring exact recall and specialized knowledge—e.g., college-level STEM disciplines like chemistry and medicine. These topics demand precise understanding, and the model's hesitation reflects its awareness of knowledge limitations.

