# OpenReview forum: "Learning to Route LLMs with Confidence Tokens"
_ICML.cc/2025/Conference — ICML 2025 poster_

### Official Review · Reviewer_JjAL · 2025-03-12

**Overall Recommendation:** 4

**Summary:**

The paper introduces Self-REF, a lightweight fine-tuning framework designed to teach large language models (LLMs) to express confidence in their answers through confidence tokens. These learned tokens, indicate whether the model is confident or uncertain about its prediction, improving reliability and performance in downstream routing and rejection learning tasks.
The authors demonstrate that Self-REF outperforms traditional approaches, such as verbalizing confidence or using token probabilities, on multiple public datasets by enabling more accurate routing to stronger models and better rejection of uncertain answers. The method achieves improved system efficiency and calibration while maintaining model performance.

## update after the rebuttal

During the rebuttal the authors have addressed all my concerns. I have decided to increase my score from 3 to 4

**Claims And Evidence:**

The claims in the submission are supported by clear and convincing evidence. The paper presents quantitative results across four public datasets (MMLU, OpenbookQA, GSM8K, MedQA), showing that Self-REF improves routing efficiency by reducing the number of queries sent to larger models while maintaining accuracy (e.g., Llama3-8B routes only 39% of queries to match Llama3-70B's performance). Additionally, ROC curves demonstrate superior rejection learning, and calibration metrics (ECE, Brier Score, Cross-Entropy) confirm that confidence tokens align well with correctness. However, while the authors discuss trade-offs in routing and rejection, they do not extensively analyze potential failure cases, such as when Self-REF misidentifies confidence leading to incorrect rejections or unnecessary escalations.

**Essential References Not Discussed:**

All the relevant references needed to understand the paper are discussed.

--

I found this paper that came out after ICLR submission deadline but it could be a nice reference to add to the related work section:
Dhananjay Ashok, Jonathan May, Language Models Can Predict Their Own Behavior

**Experimental Designs Or Analyses:**

The experimental design is generally strong. The routing experiments are well-structured, using multiple confidence thresholds to analyze trade-offs between accuracy and efficiency. The rejection learning task is also appropriately tested with artificially modified datasets where the correct answer is removed.  Also the calibration analysis (ECE, Brier Score, CE)  help measure model confidence alignment with correctness.`

On the other hand, a much stronger analysis would include a systematic investigation of failure cases to identify patterns in when and why Self-REF misclassifies confidence. While the paper provides overall accuracy and calibration metrics, it does not explore whether certain types of questions, knowledge domains, or reasoning patterns lead to systematic overconfidence or underconfidence. For instance, if the model consistently misroutes some type of quesstions questions, this could highlight fundamental limitations in its confidence estimation. Identifying such failure patterns would not only improve interpretability but also inform targeted improvements to Self-REF, such as adjusting confidence token fine-tuning strategies or incorporating adversarial training.

Also, an analyses of the relation between confidence threshold (more below in the questions section) could help get a better understanding of model behavior when self-REF is used.

With the analysis above, I consider this paper to deserve a score of 4 instead of its current 3

**Methods And Evaluation Criteria:**

The proposed method and evaluation criteria are well-aligned with the problem of confidence-based routing and rejection learning in LLMs. The use of confidence tokens is a novel yet intuitive approach that integrates seamlessly into autoregressive models, and the evaluation on standard QA and reasoning benchmarks (MMLU, OpenbookQA, GSM8K, MedQA) allows for fair comparison with prior art. The routing and rejection tasks are practical and relevant, as they reflect real-world scenarios where LLMs need to manage uncertainty efficiently.
As stated above, an error analysis of model behvior in confidence-based routing would help gain additional insights on the effectiveness of Self-REF.

**Other Comments Or Suggestions:**

Typos:

- line 117: retarded? should be a typo
- line 349: bed is s typo

**Other Strengths And Weaknesses:**

The paper is generally well written and easy to follow.

**Questions For Authors:**

- What data did you use in the setup when you mention you trained on MMLU? I looked at table 4 in the appendix but I'm not sure about the numbers given that, from the MMLU paper, I get "The few-shot development set has 5 questions per subject, the validation set may
be used for selecting hyperparameters and is made of 1540 questions, and the test set has 14079 questions.". In general, i'd move some information about training the confidence tokens from the appendix to the main body of the work, or at least better discuss them in the paper.

- In figure 2 I see random route to 70B baseline that I can't find it described. Can you elaborate on it?

- I am slightly confused by the quantile thresholds in Section 4.2 and their relation to the routing rate in Figure 2. Specifically, how do these two quantities interact? For example, in Figure 2(a), what was the exact threshold value (t) that resulted in a routing rate of 0.4? More generally, it would be helpful to explicitly discuss the relationship between t and routing rate, as this would give practitioners a clearer understanding of how to tune Self-REF for different trade-offs between accuracy and efficiency.

- Can you clarify what you mean with the following sentence about the in-context learning for llama3: "All experiments utilize Llama3-70B-Instruct with only its strong in-context learning capabilities during instance routing. The routing decisions are determined by the probabilities"

**Relation To Broader Scientific Literature:**

The paper builds on prior work in uncertainty quantification, LLM routing, and rejection learning, but introduces confidence tokens as a mechanism for end-to-end confidence estimation. Previous methods, such as logit-based calibration  and verbalized uncertainty prompts, shows worse alignment between confidence scores and correctness, whereas Self-REF fine-tunes the LLM itself to embed confidence directly. Unlike external classifiers for routing, Self-REF integrates confidence estimation within the autoregressive decoding process, making routing decisions more adaptive and model-aware. Additionally, it improves on LLM rejection learning by enabling confidence-based abstention without requiring a separate rejection model or new loss functions.

**Theoretical Claims:**

N/A

---

> ### Author Rebuttal · Authors · 2025-04-01
>
> Thank you for the valuable feedback and reference. We've incorporated them into our related work and made the suggested expository improvements in our revised paper.
>
> **[Q-1] More analysis on a systematic investigation of failure cases to identify patterns in when and why self-REF misclassifies confidence.**
>
> **[A-1]** This is a good suggestion, we previously included some case studies of successful routing in Appendix D, and will add more failure cases as well. Analyzing the top 5 categories of correct/incorrect and overconfident/underconfident predictions, we have:
> - Predict Correctly+<UN> (underconfident in its predictions): (1) computer_security; (2) high_school_biology; (3) high_school_european_history; (4) human_sexuality; (5) miscellaneous
> - Predict Wrongly+<CN> (overconfident in its predictions): (1) college_computer_science; (2) conceptual_physics; (3) high_school_computer_science; (4) high_school_microeconomics; (5) jurisprudence
> - Predict Correctly+<CN>: (1) international_law; (2) college_biology; (3) moral_disputes; (4) philosophy; (5) us_foreign_policy
> - Predict Wrongly+<UN>: (1) abstract_algebra; (2) anatomy; (3) college_chemistry; (4) college_medicine; (5) econometrics
>
> Analyzing these categories, we note:
> - Predict Correctly+<UN> (underconfident in its predictions): The model often hesitates with context-heavy subjects due to their ambiguity and need for broader reasoning. "miscellaneous" reflects general uncertainty in non-standard topics.
> - Predict Wrongly+<CN> (overconfident in its predictions): Many of these involve technical, structured domains, where the model shows overconfidence, likely due to familiarity from training. However, it may struggle with edge cases and nuanced reasoning, especially in areas like jurisprudence.
> - Predict Correctly+<CN>: These topics rely on broad conceptual knowledge rather than strict calculations, and the model appears well-calibrated, likely due to strong training data coverage or clearer signals of correctness.
> - Predict Wrongly+<UN>: These highly specialized, detail-heavy subjects require precise recall or deep understanding, and the model’s uncertainty may reflect an awareness of its limitations in recalling detail-heavy content to answer questions.
>
> **[Q-2] What data did you use in the setup when you mention you trained on MMLU? In general, i'd move some information about training the confidence tokens ...**
>
> **[A-2]** We use a randomly sampled subset of the official MMLU training set, where ground truth answer choices are augmented with confidence tokens when fine-tuning Self-REF on MMLU (Algorithm 1, Section 3.2). We will move more details about the training setup from the appendix into section 4.1.
>
> **[Q-3] In figure 2 I see random route to 70B baseline that I can't find it described. Can you elaborate on it?**
>
> **[A-3]** The random routing approach is a naive baseline that uniformly at random routes to the 70B model at the specified rate. When the random routing rate is 0.0, then it gives the performance of the small LM, and when the random routing rate is 1.0, it gives the performance of routing to the 70B model entirely. We will update our paper with these details.
>
> **[Q-4] I am slightly confused by the quantile thresholds in Section 4.2 and their relation to the routing rate in Figure 2. Specifically, how do these two quantities interact? for example, in Figure 2(a), ...? More generally, it would be helpful to explicitly discuss the relationship between t and routing rate, as ... trade-offs between accuracy and efficiency.**
>
> **[A-4]** To better analyze the routing performance, we set the thresholds t at 20-quantiles, as described in Section 4.2. For instance, the threshold t corresponding to a routing rate of 0.4 is the 40th percentile of confidence scores across all input queries. Practically, one could observe the 20-quantiles of the distribution, and choose thresholds that should empirically correspond to rough routing rates, and then monitor these thresholds over time. Note that Self-REF fine-tuning does not require committing to a particular tradeoff, as instead of relying directly on the sampled tokens, we instead extract confidence scores from the logits of the confidence tokens, thus giving us the ability to threshold this score and control how often to route.
>
> **[Q-5] Can you clarify what you mean with the following sentence about the in-context learning for llama3: "All experiments utilize Llama3-70B-Instruct with only its strong in-context learning capabilities during instance routing..."**
>
> **[A-5]** This refers to the fact that we did not fine-tune Llama3-70B-Instruct (the larger, more expensive LLM) in the routing setting.
>
> **[Q-6] Missing reference, formats, and Typos.**
>
> **[A-6]** Thank you for the valuable comments and references. We will add this to our related work section on rejection learning. The suggestions for formats and typos are fixed in our next version.

---

> > ### Comment · Reviewer_JjAL · 2025-04-02
> >
> > Thanks for your response. Thing are clearer now. I don't see any major weakness with this work and have decided to increase my score to 4

---

### Official Review · Reviewer_gYcY · 2025-03-17

**Overall Recommendation:** 3

**Summary:**

Self-REF is a lightweight method for training a language model to show when its answers are correct or incorrect by using “confidence tokens.” The approach starts with a base model that generates predictions and labels each instance as “confident” or “unconfident” based on the answer’s correctness, creating an augmented dataset. The model is then fine-tuned on these labeled samples and learns both to provide the correct answer and to tag it appropriately. Finally, continuous confidence scores are computed by comparing the probabilities of the “confident” vs. “unconfident” token at the end of each response.

They propose two applications. First, low-confidence queries can be routed to a larger model to cut costs without sacrificing accuracy. Second, when no larger model is available, the system can reject responses it deems untrustworthy. Across MMLU, OpenbookQA, GSM8K, and MedQA, routing only uncertain queries matches the strong model’s accuracy while reducing latency and cost; rejecting low-confidence outputs also helps avoid incorrect claims for questions lacking a valid answer.

**Claims And Evidence:**

* Claim that the model’s confidence tokens always align with correctness:
The paper states that once fine-tuned, the model should consistently output <CN> whenever it is correct and <UN> otherwise. In practice, even well-calibrated methods can yield discrepancies between predicted confidence and true correctness.

* Claim of consistently “lightweight” overhead
The resulting method would yield a cascading method, where first we have to monitor the smaller model's output then route to the bigger one to get the final answer.

**Essential References Not Discussed:**

NA

**Experimental Designs Or Analyses:**

Yes, I would suggest studying additional baselines:
* I'm curious if authors explored a baseline where the model generates a confidence token before producing an answer. This can reduce computation by preventing full generation when confidence is low. Also a learned router baseline would provide insight
* Explore other methods for labeling confidence:
    * Estimating confidence based on consistency across multiple stochastic samples from the model.
    * External calibration techniques, e.g. entropy-based uncertainty

**Methods And Evaluation Criteria:**

Overall, the methods and evaluation criteria do generally make sense given their focus on practical improvements in downstream tasks. However, I have a few concerns:
* The assumption that <CN> tokens represent correctness and <UN> represent incorrectness lacks a clear theoretical grounding. The paper implicitly assumes confidence directly correlates with correctness, though confidence in practice might not always reflect correctness accurately, potentially limiting generalizability.
* The paper partially addresses imbalance by subsampling unconfident tokens with a tunable parameter, but does not elaborate on handling extreme cases where the base model might be consistently correct or incorrect, which could severely skew training data. In fact, the approach depends heavily on the correctness of the base model to annotate confidence tokens. If the base model has substantial weaknesses or biases, these may propagate through fine-tuning, limiting improvements or perpetuating biases.

**Other Comments Or Suggestions:**

NA

**Other Strengths And Weaknesses:**

NA

**Questions For Authors:**

NA

**Relation To Broader Scientific Literature:**

The work addresses an important problem of allowing LLMs to self-assess the confidence of their predictions, and their applications to LLM routing and rejections.

**Theoretical Claims:**

NA

---

> ### Author Rebuttal · Authors · 2025-04-01
>
> Thank you for the detailed feedback. We respond to each point below, and will update the discussion accordingly:
>
> **[Q-1] The assumption that <CN> tokens represent correctness and <UN> represent incorrectness lacks a clear theoretical grounding. The paper implicitly assumes confidence directly correlates with correctness...**
>
> **[A-1]** A core contribution of this paper is utilizing confidence scores for rejection and routing downstream. Unlike prior works that derive uncertainty scores from logits, verbalized outputs, or re-sampling, we train our notion of confidence into the model, teaching it to reflect on its answer after producing it. The notion of confidence is grounded in the probability that a prediction is correct, i.e., $P(\text{<CN>} |X,Z) = P(Y=Z|X,Z)$, where CN is the confidence token, $X$ is the question, $Z$ is the predicted answer, and $Y$ is the true answer. This is a more direct form of confidence than other techniques, which often instead reflect consistency in the answer. For example, logits would instead reflect $P(Z|X)$, with no relation to $Y$. As noted in our paper (Section 1 and 5.3), calibrated uncertainty metrics are not necessarily correlated with correctness. In our paper, we assess both the utility of the confidence scores as well as their calibration.
>
> **[Q-2] If the base model has substantial weaknesses or biases, these may propagate through fine-tuning, limiting improvements or perpetuating biases.**
>
> **[A-2]** We respond using two perspectives:
> - The primary goal of Self-REF is to help the base model identify its existing weaknesses while maintaining performance, thereby enabling more effective routing.
> - One possible approach for extreme incorrectness is as follows: first, fine-tune the base model on the downstream task to improve its task-specific capabilities; then, in a second stage, apply the Self-REF framework to teach the model to route effectively based on uncertainty. This is a useful extension of Self-REF we will add to the discussion.
>
> **[Q-3] Explore other methods for labeling confidence (consistency across sampling and external calibration).**
>
> **[A-3]**  As mentioned in Q-1, the goal of Self-REF is to produce confidence tokens useful for downstream settings such as routing and rejection, boosting correctness of the overall system. Well-calibrated confidence scores do not always correlate with correctness [1, 2] (Section 1 and 5.3). Two toy examples to explain this misalignment intuitively: Assume we have a binary classification task with predicted probabilities (see below). Example 1 achieves lower accuracy but has a lower ECE score, whereas Example 2 achieves higher accuracy but a higher ECE score. This example demonstrates that calibration metrics are not correlated with the correctness of the prediction. Similarly, consistency-based uncertainty is not necessarily aligned with downstream correctness, e.g., when one has highly consistent incorrect answers. This misalignment can degrade routing performance when such signals are used for routing.
>
> ```
> Bins = 2
> predicted prob. of "1" = [0.5, 0.5, 0.5, 0.5]
> ground truth = [0, 0, 1, 1]
>
> Example 1:
> predicted prob. of "1" = [0.5, 0.5, 0.5, 0.5]
> -> ECE(↓)= 0%
> -> accuracy (↑)= 0%
>
> Example 2:
> predicted prob. of "1" = [0.4, 0.6, 0.9, 0.9]
> -> ECE(↓) = 20%
> -> accuracy (↑) = 75%
> ```
> [1] Huang, et al. "Look before you leap: An exploratory ..." arXiv 2023.
>
> [2] Yona, et al. "Can Large Language ... Uncertainty in Words?." arXiv 2024.
>
> **[Q-4] I'm curious if authors explored a baseline where the model generates a confidence token before producing an answer. Also a learned router baseline would provide insight.**
>
> **[A-4]** To the best of our knowledge, there is no existing baseline that generates a confidence token prior to producing the answer. In multiple-choice QA, token generations are relatively short, making it feasible to predict a confidence token after producing the answer. This allows the model to condition its confidence estimation directly on the generated output, potentially resulting in more accurate confidence scores for routing. However, for tasks involving longer responses (e.g. reasoning), an alternative approach to improve efficiency could involve an early-stopping mechanism for confidence tokens. The model might produce a confidence token midway through its answer generation, allowing an earlier routing decision.
>
> We have additional experiments with a learned confidence-based router baseline: OOD-Probe [1] to assess its routing performance from Mistral-7B to Llama3-70B in the MMLU dataset. The results are shown in the table below. We provide the accuracy in different routing ratios, and observe that Self-REF outperforms the new baseline.
>
> |Route_Ratio|0%|20%|40%|60%|80%|100%|least_routing_ratio|
> |-|-|-|-|-|-|-|-|
> |OOD-Probe|0.50|0.61|0.66|0.68|0.74|0.74|80%|
> |Self-REF|0.55|0.64|0.68|0.72|0.74|0.74|70%|
>
> [1] Mahaut, et al. "Factual confidence of LLMs:..." ACL 2024.

---

### Official Review · Reviewer_w1rW · 2025-03-17

**Overall Recommendation:** 4

**Summary:**

Authors proposed a lightweight training strategy to teach LLMs to express confidence in whether their answers are correct in a reliable manner. Using this, the authors build a router algorithm that reduces latency and improves overall QA performance.

**Claims And Evidence:**

The claims are well stated and supported.

**Essential References Not Discussed:**

None

**Experimental Designs Or Analyses:**

Experimental designs and analyses are valid. In particular, significantly reducing the latency metric, improves the validity of the results provided.

**Methods And Evaluation Criteria:**

All three RQ1, RQ2, RQ3 benchmarks and datasets are make sense and well analyzed.

**Other Comments Or Suggestions:**

Paper is well written.

**Other Strengths And Weaknesses:**

Please see other sections

**Questions For Authors:**

Your framework is only able to predict the confidence token at the end of the answer? Is that correct?
If so, I see the drawback in terms of routing capabilities. since the model need to generate the full response before going to route the response to another larger LLM? Do authors see any way to get the confidence tokens before the full answer is generated, using the proposed framework?

**Relation To Broader Scientific Literature:**

The routing problem and Confidence Tokens are very novel research area. Every related literature is well covered.

**Theoretical Claims:**

Not applicable

---

> ### Author Rebuttal · Authors · 2025-04-01
>
> Thank you for the valuable feedback and ideas. We have updated our discussion accordingly.
>
> **[Q-1] Your framework is only able to predict the confidence token at the end of the answer? Is that correct? If so, I see the drawback in terms of routing capabilities. Since the model need to generate the full response before going to route the response to another larger LLM? Do authors see any way to get the confidence tokens before the full answer is generated using the proposed framework?**
>
> **[A-1]** Thank you for the insightful feedback. In question-answering tasks, the cost of generation is relatively low, and predicting a confidence token after generating the answer allows the model to condition its confidence on the output, potentially leading to more accurate uncertainty estimation for routing. However, in other long-response tasks, such as LLM-based reasoning, a promising direction could be to develop an early-stopping approach for confidence token generation. The model may emit a confidence token partway through answer generation in this setup, enabling an earlier routing decision to stronger models. This strategy could be particularly beneficial in complex reasoning scenarios, and we consider it a valuable direction for future work.

---

### Official Review · Reviewer_kmXF · 2025-03-19

**Overall Recommendation:** 3

**Summary:**

The paper proposes Self-REF, a training strategy that adopts LoRA to fine-tune an LM on a dataset augmented with confidence tokens, based on prediction correctness. Self-REF enhances downstream applications like model routing and answer rejection by leveraging the learned confidence token scores.

**Claims And Evidence:**

Please refer to the "Other Strengths And Weaknesses" section.

**Essential References Not Discussed:**

Please refer to the "Other Strengths And Weaknesses" section.

**Experimental Designs Or Analyses:**

Please refer to the "Other Strengths And Weaknesses" section.

**Methods And Evaluation Criteria:**

Please refer to the "Other Strengths And Weaknesses" section.

**Other Comments Or Suggestions:**

Presentation: The figure size and font size (e.g., in ```Figure 2```) might be too small, potentially affecting readability.

**Other Strengths And Weaknesses:**

**Strengths**
- The proposed Self-REF method is straightforward yet effective in learning more calibrated confidence scores, utilizing a simple data augmentation strategy.

- Overall, the paper is well-written and easy to follow, with comprehensive and detailed experiments that demonstrate strong empirical results across multiple datasets.

**Weaknesses**
- Missing reference to R-Tuning [1], which similarly constructs an augmented dataset consisting of certain and uncertain sets based on the correctness of LM predictions, and appends corresponding tokens as supervised signals to fine-tune a more calibrated LM capable of refraining from answering unknown questions. This work is not discussed in the paper, and could be considered as a baseline.

- Self-REF requires a dedicated training/validation set and a fine-tuning stage for different downstream tasks/datasets, which limits its practical usages comparing to zero-shot baselines.

- It is unclear how much transferability Self-REF holds beyond similar tasks such as OpenbookQA --> MMLU. It would be very interesting to see how Self-REF performs when fine-tuning in a multi-task setup with a collective of datasets.

[1] Zhang, Hanning, et al. *R-tuning: Instructing large language models to say ‘I don’t know’.* NAACL 2024

**Questions For Authors:**

1. How was the optimal parameter $\alpha$ for the <UN> data determined?
2. How does the ratio of <CN>:<UN> in dataset construction affect the results?
3. In ```Figure 2```: why do different models/baselines/datasets have different number of datapoints? For instance, Mistral only has 3 data points for the "Verbalizing uncertainty" baseline curve on MMLU.

**Relation To Broader Scientific Literature:**

Please refer to the "Other Strengths And Weaknesses" section.

**Theoretical Claims:**

N/A. No proofs or theoretical claims that require checking.

---

> ### Author Rebuttal · Authors · 2025-04-01
>
> Thank you for the valuable feedback and reference. We've incorporated them into our related work and made the suggested expository improvements in our revised paper.
>
> **[Q-1] Self-REF requires a dedicated training/validation set and a fine-tuning stage for different downstream tasks/datasets, which limits its practical usages compared to zero-shot baselines.**
>
> **[A-1]** Our method targets the scenario where one would like to optimize performance for a particular downstream task while (1) being able to fine-tune smaller specialized models, (2) reducing monetary or latency cost, and (3) maintaining comparable performance on the downstream task.
> Thus, we agree that fine-tuning is necessary in self-REF. However, we argue that it is often the case that one may want to fine-tune a smaller model for one's specific task of interest. Exploring broader use cases is a valuable direction for future work. An interesting extension would be to explore how self-REF can extend to multi-task training for general use cases.
>
> **[Q-2] It is unclear how much transferability Self-REF holds beyond similar tasks such as OpenbookQA --> MMLU. It would be very interesting to see how Self-REF performs when fine-tuning in a multi-task setup with a collective of datasets.**
>
> **[A-2]** We agree that fine-tuning self-REF with a mixture of tasks augmented with confidence tokens is a promising extension. However, we would like to emphasize that the nature of uncertainty can vary significantly across tasks. For instance, uncertainty in question-answering may differ from that in code generation or other complex reasoning tasks. While QA tasks often have well-defined ground truth, other tasks may require soft confidence labels (tokens), which could affect the effectiveness of confidence-based routing. We value and acknowledge the potential of this direction and will consider incorporating a broader set of datasets under a multi-task framework in future work.
>
> **[Q-3] How was the optimal parameter $\alpha$ for the <UN> data determined?**
>
> **[A-3]** $\alpha$ is a hyperparameter in our work, and we select the optimal $\alpha$ based on the validation set.
>
> **[Q-4] How does the ratio of <CN>:<UN> in dataset construction affect the results?**
>
> **[A-4]** We fixed the overall dataset size and treated the ratio between <CN> and <UN> samples as a hyperparameter to optimize performance for a particular downstream task with fine-tuning. On one hand, including more <CN> samples helped fine-tune the model to better perform the downstream task; on the other hand, a sufficient number of <UN> samples was necessary to teach the model when to express uncertainty. An imbalance—either too many <UN> samples, which could degrade task performance, or too many <CN> samples, which could reduce the quality of uncertainty estimation—can negatively affect the model. To balance this trade-off, we experimented with ratios of 1:1, 1:2, 1:3, 1:4, and 1:5 and selected the optimal setting based on each task’s validation set.
>
> **[Q-5] In Figure 2: why do different models/baselines/datasets have different number of datapoints? For instance, Mistral only has 3 data points for the "Verbalizing uncertainty" baseline curve on MMLU.**
>
> **[A-5]** The points correspond to 20-quantiles, as described in Section 4.2, which ideally yield 20 distinct and equally spaced routing rates, assuming all confidence scores are unique. However, certain methods for extracting confidence scores, such as verbalizing uncertainty, can suffer from mode collapse, where the scores cluster around a limited set of numbers. As a result, a substantial portion of the confidence scores is identical, leading to fewer than 20 distinct routing rates in practice.
>
> **[Q-6] Missing reference, formats, and Typos.**
>
> **[A-6]** Thank you for the valuable comments and references. We will add this to our related work section on rejection learning. The suggestions for formats and typos are fixed in our next version.

---

### Decision · Program_Chairs · 2025-05-01

**Decision:**

Accept (poster)

**Comment:**

**Paper summary:**

This paper proposes a framework to fine-tune an LLM to express its confidence for an input query through a custom confidence token that is annotated in the fine-tuning data set. The framework has three phases:

1. Confidence token annotation. If the model gives a correct answer, append <CN> to the ground-truth response. Else, append <UN>.
2. LoRA fine-tuning. Fine-tune the model on the augmented data.
3. (Inference time) Confidence score extraction. After decoding, the probability of <CN> (appropriately normalized) is a proxy for model confidence.

Two applications are considered: 1. learning to reject, 2. model cascading.

**Reviews**

The reviewers noted that the proposed approach is simple and yet effective. The paper is well written and easy to follow (kmXF, w1rW). At a high level, there are two main claims: 1. confidence tokens align with correctness, 2. small overhead. These claims are well supported by the provided empirical results  (gYcY, JjAL). Experiments are thorough.

Most of the questions raised were mainly clarification questions  (e.g., ratio of <CN> and <UN>), to which the rebuttal addressed them well. Among the remaining concerns, as may be expected, a number of reviewers (w1rW, gYcY) asked why the special confidence token is appended at the end (as opposed to at the beginning, which can save some decoding time). While the rebuttal states that  conditioning on the generated response can help improve uncertainty quantification, no further results were provided. That the work does not extensively analyze potential failure cases (JjAL) is another concern. In the rebuttal, the authors listed a few instances from a few tasks where the method tends to fail, partially addressing this concern. It would be good if the authors could please analyze failure cases in more detail and provide general insights (as opposed to a case-by-case analysis).  Owing to the extensive and strong empirical results, the AC thinks the merits outweigh the drawbacks.

AC recommendation: accept.